# Preliminary study of microbiologically influenced corrosion by Pseudomonas aeruginosa on high Chromium white iron

Cedric Tan*, Naveen Kumar Elumalai*, Kannoorpatti Narayanan Krishnan

Faculty of Science and Technology, Energy and Resources Institute, Advanced Manufacturing Alliance, Charles Darwin University, Darwin, Northern Territory, Australia

* cedric.tan@cdu.edu.au (CT); naveenkumar.elumalai@cdu.edu.au (NKE)

**Data Availability Statement:** All relevant data are within the paper and its Supporting information.

**Funding:** Australian Government RTP funding was provided as scholarship to Cedric Tan

## Abstract

Microbiologically Influenced Corrosion (MIC) poses a significant challenge to various industries, leading to substantial economic losses and potential safety hazards. Despite extensive research on the MIC resistance of various materials, there is a lack of studies focusing on High Chromium White Iron (HCWI) alloys, which are widely used in wear-resistant applications. This study addresses this knowledge gap by providing a comprehensive investigation of the MIC resistance of three HCWI alloys with varying chromium contents (22 wt%, 30.7 wt%, and 21 wt%) in the presence of Pseudomonas aeruginosa (*P. Aeruginosa*), a common bacterial species associated with MIC. The alloys were exposed to an artificial seawater medium inoculated with *P.Aeruginosa* for 14 days, and their corrosion behaviour was evaluated using electrochemical techniques, surface analysis, and microscopy. Electrochemical Impedance Spectroscopy (EIS) results revealed that the alloy with the highest chromium content (A2, 30.7 wt% Cr) exhibited superior MIC resistance compared to the other alloys (A1, 22 wt% Cr and M1, 21 wt% Cr). The enhanced performance of alloy A2 was attributed to the formation of a more stable and protective passive film, as well as the development of a more compact and less permeable biofilm. The EIS data, interpreted using equivalent circuit models, showed that alloy A2 had the highest charge transfer resistance and the lowest biofilm capacitance, indicating a more effective barrier against corrosive species. Bode plots further confirmed the superior corrosion resistance of alloy A2, with higher impedance values and phase angles at low frequencies compared to alloys A1 and M1. Scanning Electron Microscopy (SEM) and optical microscopy analyses corroborated these findings, showing that alloy A2 had the lowest pit density and size after 14 days of exposure. The insights gained from this study highlight the critical role of chromium content in the MIC resistance of HCWI alloys and have significant implications for the design and selection of materials for applications prone to microbial corrosion.

**Competing interests:** The authors have declared that no competing interests exist.

## 1. Introduction

High chromium white irons are used extensively in mining and metallurgy industries and in applications requiring wear resistance. One of the properties expected of these alloys is hardness [1]. Though the alloys are used in environments with a range of pH values, there is little information on the corrosion and Microbiologically Influenced Corrosion (MIC) behaviour of these alloys. These alloys can be applied as coatings onto metals via weld deposition to improve wear resistance [1].

Corrosion is estimated to cost up to 4% of the annual global GDP, with MIC itself being responsible for approximately 20% of all corrosion related costs [2]. MIC is corrosion is caused by the exposure of metals to bacteria, forming a biofilm layer and subsequently leading to localised pits and crevices on the surface of materials. Javaherdashti argues that the term biofilm in MIC be replaced with the more suitable term "temenos" to emphasise upon the fact that under-'biofilm' conditions are far different from those of the bulk solution [3]. However, due to currently accepted usage and familiarity among the scientific community involved in MIC research, these layers are still referred to as "biofilms" in this paper. Specific mechanisms for MIC vary per organism, though in general can be described as either producing corrosive byproducts or inhibiting corrosion resistance [4]. Mining and oil and gas industries have to deal with environments with a range of pH values. There are also other applications where hardfacing alloys are used with a range of pH and the environments encountered in sugarcane industry have a pH of about 3 [5] and in the alumina industry the pH is 14 [6]. Tungsten carbides used in the drill bits of mining have been found to corrode [7] and in cemented carbides [8] due to low pH encountered in drilling.

MIC occurs in piping and equipment of the infrastructure across the oil and gas networks and corrosion failures were about 25% of all failures [9]. One MIC causing species of interest is that of Pseudomonas Aeruginosa (*P.Aeruginosa*), which can be found dominantly in marine environments [10]. *P.Aeruginosa* was chosen in this study due to being commonly found [10,11]. As a gram-negative bacteria, the cell wall of *P.Aeruginosa* is negatively charged, contributing towards its adhesion on metal surfaces. This adhesion ability is fundamental towards the initial phase of surface biofilm formation [11]. Other factors that can affect adhesion ability include, but are not limited to, surface roughness, material hydrophobicity/hydrophilicity, and environmental composition [12]. Even so, it should be cautioned that adhesion should not necessarily be taken as a direct link to corrosion rate when comparing different alloys given findings such as for *P.Aeruginosa* with stainless steels. For instance, 316 stainless steel has been seen to have a higher adhesion to *P.Aeruginosa*, but with lower corrosion rate than 201 and 304 stainless steel. [13]

Chromium content in HCWI is desirable against corrosion due to its ability to form passive films [14]. However, this film formation ability is affected under the influence of Pseudomonas Aeruginosa, especially due to synergistic attack from chloride ions being concentrated by the bacteria [11,15,16]. It could be suggested that increasing chromium content may be attempted as a countermeasure to maintain protective film stability. Additionally, released chromium ions during corrosion would increase its local concentration around the surface of alloys. This may lead to improved protection as released chromium ions become incorporated into protective chromium oxide films, as demonstrated for abiotic supercritical conditions [17].

Finally, while studies exist for the corrosion of various stainless steel alloys against *P.Aeruginosa* [18–20], few seem to exist for alloys with high chromium and contents, as with High Chromium White Iron alloys. In this investigation 3 different alloys used in the mining industry that produce different hardness by varying chromium, carbon, and boron were taken up

for MIC study. These alloys also produce microstructures of austenite and martensite (with boron additions).

## 2. Materials and methods

### 2.1 Materials

Three different alloys were analysed; one martensitic (Alloy M1, classification AS/NZS 2576:2560-A4), and two austenitic (Alloys A1 classification AS/NZS 2576:2460-A4 and Alloy A2, classification AS/NZS 2576:2360-A4). AS/NZS 2576 specifies hardfacing alloys based on the matrix (austenite-eutectic carbide or martensite-eutectic carbide) when they are deposited by welding [1]. These alloys were primarily chosen for their high amount of chromium content. The composition of each alloy as tested using ICP-AES is provided in Table 1.

Alloys were deposited from electrodes using Shielded Metal Arc Welding onto mild steel, using a minimum four-layer deposition method to avoid dilution, as described in AS/NZS:2576:2005 [1]. The substrate was then removed by a precision cutting machine and cut to expose a flat 1cmx1cm square surface, as indicated in Fig 1. This hardfacing sample was then affixed to a conductor and mounted in epoxy for use as working electrodes, as shown in Fig 2. Before testing, all samples were polished to 1μm. Electrodes were then ultrasonically washed with acetone for 10 minutes, immersed within 80% ethanol for 1 hour, and finally rinsed with high purity water to reduce the chance of bacterial contamination on the surface.

### 2.2 Bacteria and test medium

Artificial seawater was formulated as per ASTM D1141-09, with the composition described in Table 2. [21]

Pseudomonas Aeruginosa (ATTC BAA-1744) was cultivated from glycerol stock stored at -80˚C. This strain can be described to be rod shaped and from a clinical isolate stock [22]. This was streaked onto Tryptic Soy Agar petri dishes and left to culture overnight in a 37˚C incubator. Bacteria was then transferred to 75 mL of Tryptic Soy Broth (TSB) for further culturing over an additional 24 hours, as described in the handling procedure for the stock [22]. The TSB medium with *P.Aeruginosa* was added to 750mL of post-autoclaved nutrient rich artificial seawater, and finally pH adjusted to 7.4. This final solution was used as the main testing medium.

### 2.3 Electrochemical test procedure

Electrochemical testing was conducted using a three-electrode cell with an Ag/AgCl reference electrode and a platinum coated counter electrode. The artificial seawater with *P.Aeruginosa* was used as the electrolyte/test medium. All electrochemical tests were conducted using a Gamry 3000 potentiostat, with results analysed using the Gamry Echem Analyst software and ZSimpwin. Before immersing electrodes in the medium, all electrodes were ultrasonically

**Table 1. Alloy compositions as tested via ICPAES.**

| Alloy | Chemical Elements (wt%) | | | | | | | | |
|---|---|---|---|---|---|---|---|---|---|
| | Cr | C | Mn | Si | V | B | Ni | Mo | Fe |
| A1 | 22 | 2.78 | 2.11 | | 0.15 | - | 0.50 | 1.26 | Balanced |
| A2 | 30.7 | 4.52 | 1.87 | 1.43 | 0.05 | - | 0.17 | 0.16 | Balanced |
| M1 | 21 | 5.15 | 0.36 | 2.23 | 1.80% | 0.67 | 0.18 | 0.12 | Balanced |

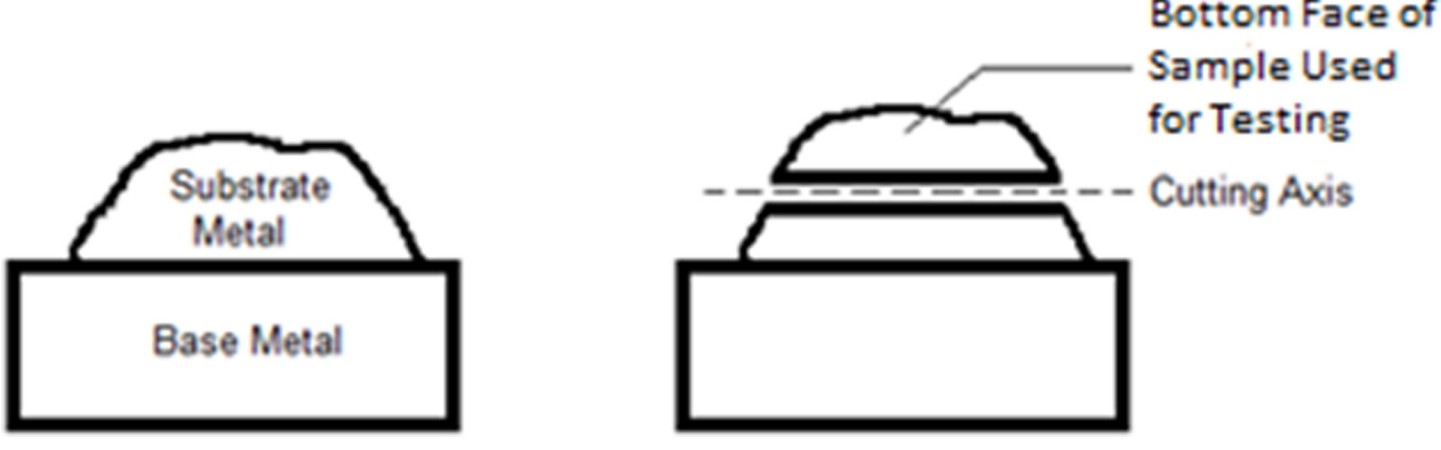

**Fig 1. Alloy cut used for testing.**

washed with acetone for 10 minutes. They were then subsequently immersed in 80% ethanol for 2 hours before rinsing with high purity water and left to dry in a biohazard cabinet.

Electrochemical Impedance Spectroscopy (EIS) was carried out for each sample on Day 7 and Day 14 at Open Circuit Potential. Prior to each EIS measurement, Open Circuit Potential (OCP) was taken after a runtime of 600 seconds. A control sample without introducing *P.Aeruginosa* to the artificial seawater medium was also conducted in parallel, comparing the EIS measurement after Day 14. The amplitude value used was 20mV, along with a frequency range of 100 mHz to 100kHz. pH was measured with a pH meter (TPS pH Cube) every 2 days of testing.

## 2.4 Adhesion test procedure

Fluorescence microscopy was used to measure adhesion of bacteria onto the material surfaces. To prepare the samples, coupons of each alloy (ie. A1, A2, and M1) were placed polished-side-

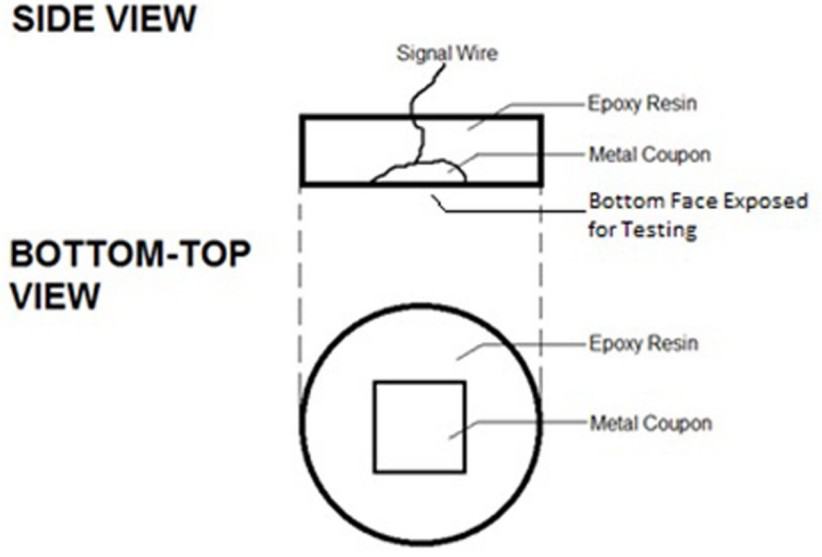

**Fig 2. Electrode prepared from cut metal sample.**

**Table 2. Composition of artificial seawater.**

| Compound | Concentration (g/L) |
|---|---|
| NaCl | 24.53 |
| MgCl$_2$ | 5.20 |
| Na$_2$SO$_4$ | 4.09 |
| CaCl$_2$ | 1.16 |
| KCl | 0.695 |
| NaHCO$_3$ | 0.201 |
| KBr | 0.101 |
| H$_3$BO$_3$ | 0.027 |
| SrCl$_2$ | 0.025 |
| NaF | 0.03 |

up within 50mL beakers, and then filled with the *P.Aeruginosa* in artificial seawater medium. Parafilm was used to cover each beaker, and subsequently left in a 37˚C incubator overnight. After incubation, alloys were rinsed with 1x Phosphate Buffered Saline (PBS) three times to remove poorly adhered bacteria. Cells were fixed with the addition of 4% formaldehyde in PBS over 20 minutes and rinsed again with 1x PBS three more times. Finally, 300nM 4',6-diami-dino-2-phenylindole (DAPI) was used to stain the solution, requiring the samples to be protected from light over 10 minutes before rinsing with 1x PBS three more times. Using fluorescence microscopy, stained cells were counted for 10 randomly selected fields on each surface. The overall test procedure was repeated three times.

## 2.5 Surface analysis

Microstructure of as-deposited samples was observed using a Nikon Eclipse MA100 microscope, after etching the samples with Nital.

Scanning Electron Microscope (Phillips XL 30) images were taken to observe alloy microstructure and surface bacteria. SEM imaging was completed for samples on Day 7 and Day 14. This was initially prepared by removing the samples from the solution, rinsing with 1x PBS three times, and subsequently fixed with 4.5% glutaraldehyde in PBS for 30 minutes. Samples were then washed with high purity water three times, before being subject to progressive ethanol drying to prevent microstructural collapse. This was achieved via stepwise dehydration with 25%, 50%, 75%, 90%, and 100% ethanol, with 10 minutes being taken for each step. A final 30 minutes was then taken to ensure complete drying in a biohazard cabinet. Chemical analysis was conducted for the samples immersed for 14 days using energy-dispersive x-ray spectroscopy (EDS). For observation of the microstructure without biofilm, the 14-day samples were washed with high purity water three times and cleaned with an ultrasonic bath for 10 minutes. The surface was cleaned by immersing in Clarke's solution, as per ASTM G1-03 [23]. This was rinsed three more times with high purity water before observation under a microscope.

## 3. Results

### 3.1 pH results

Fig 3 shows the change in the pH of the solution over the two-week testing period for each of the three HCWI alloys (A1, A2, and M1) exposed to Pseudomonas aeruginosa in an artificial seawater medium.

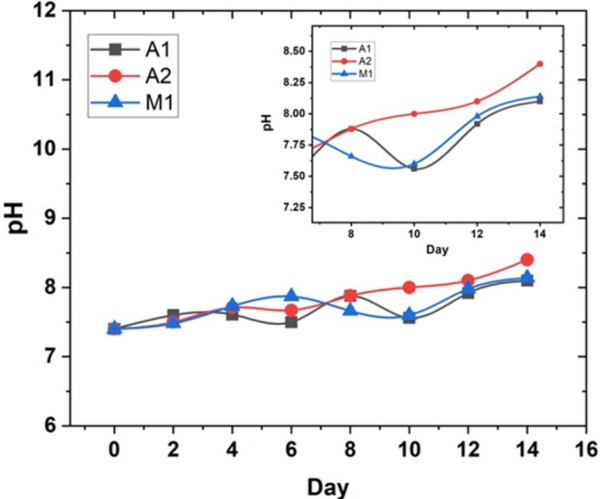

**Fig 3. Change in solution pH over time (with insert at higher resolution).**

The key observation from Fig 3. is that the pH of the solution increased for all three alloys over the course of the experiment. The pH started at around 7.4 for all alloys and increased to around 8.1 for alloys A1 and M1, and 8.4 for alloy A2 by the end of the two-week period. The primary mechanism of corrosion in this system is likely due to the formation of differential aeration cells caused by the bacteria biofilm [11].

In a differential aeration cell, the area beneath the biofilm acts as an anodic region due to limited oxygen availability, while the exposed surface acts as a cathodic region with abundant oxygen. The anodic and cathodic regions form a corrosion cell, leading to the dissolution of alloy from the metal surface and the reduction of oxygen at the cathodic sites (cathodic reaction: $O_2 + 2H_2O + 4e^- \rightarrow 4OH^-$). The cathodic reaction produces hydroxide ions ($OH^-$), which are released into the bulk solution, causing an increase in the solution's pH over time. This mechanism explains the observed trend of increasing pH in Fig 3. as the corrosion process progresses and more hydroxide ions are generated from the cathodic reaction.

### 3.2 OCP results

The Open Circuit Potential (OCP) results presented in Fig 4. provide valuable insights into the corrosion behaviour of the three HCWI alloys (A1, A2, and M1) in the presence of *P. aeruginosa*.

After one day of immersion, all three alloys exhibited a rapid drop in OCP, with alloy A2 showing the least negative potential. This initial decrease in OCP can be attributed to the quick formation of a biofilm on the alloy surfaces, which alters the electrochemical properties of the metal-solution interface [24]. The less negative OCP of alloy A2 suggests a slower biofilm formation rate compared to alloys A1 and M1, which could be due to its higher chromium content (30.7 wt% Cr) promoting the formation of a more stable and protective passive film [25,26].

After the initial drop, the OCP values stabilized for all alloys during the first week of the test period. This stabilization indicates that the biofilms reached a steady state, and the corrosion process was likely controlled by the diffusion of species through the biofilm [19].

Towards the end of the 14-day test, alloys A1 and M1 exhibited a slight increase in OCP compared to their values on Day 1–2. This increase could be attributed to the onset of localized

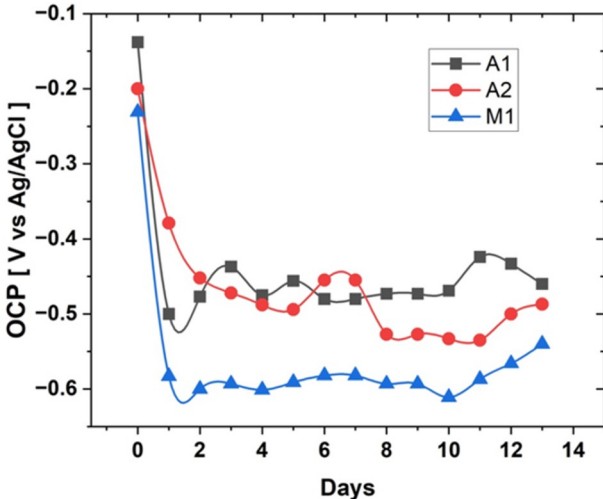

**Fig 4. OCP data over 14 days.**

corrosion, such as pitting, or the thinning of the biofilm on the alloy surfaces, which can lead to increased ion transfer and a higher OCP [27]. In contrast, alloy A2 showed a gradual decrease in OCP until approximately Day 11, followed by a slight increase. This behaviour suggests that the biofilm on alloy A2 continued to grow until around Day 11 before starting to thin out, indicating a more stable and protective biofilm formation on this alloy.

### 3.3 Adhesion results

Fig 5. shows the *P.Aeruginosa* count using fluorescence microscopy after 24 hours immersion in the test medium. Alloys A1 and A2, both having an austenitic microstructure, exhibited similar cell densities of approximately $17 \times 10^6$ cells/cm$^2$, with overlapping error bars indicating no significant difference between the two alloys. This similarity in bacterial adhesion suggests that the difference in chromium content between alloys A1 (22 wt% Cr) and A2 (30.7 wt% Cr) did not significantly influence the initial attachment of *P.Aeruginosa* cells to the alloy surfaces.

In contrast, alloy M1, which has a martensitic microstructure, showed a significantly higher cell density of approximately $32 \times 10^6$ cells/cm$^2$, nearly twice that of the austenitic alloys. The higher bacterial adhesion on alloy M1 can be attributed to its martensitic microstructure, which is characterized by a higher density of surface dislocations and defects compared to the austenitic microstructure [28]. These surface irregularities can act as preferential sites for bacterial attachment, promoting higher cell adhesion [28]. It is also likely that the greater adhesion rate was due to microstructure, as A1 and A2 are austenitic, whilst M1 has a martensitic matrix due to the presence of boron in its composition. [1]

Despite the higher initial bacterial adhesion on alloy M1, it is essential to consider that adhesion is just one aspect of the overall MIC process. The long-term corrosion resistance of the alloys depends on various factors, including the stability of the passive film, the nature of the biofilm, and the alloy composition [18].

Furthermore, the nature of the biofilm formed on the alloy surfaces can significantly influence the corrosion behaviour. A more compact and adherent biofilm can limit the diffusion of oxygen and other species to the metal surface, thereby reducing the corrosion rate [29]. The lower bacterial adhesion on alloy A2 may lead to the formation of a thinner and more compact

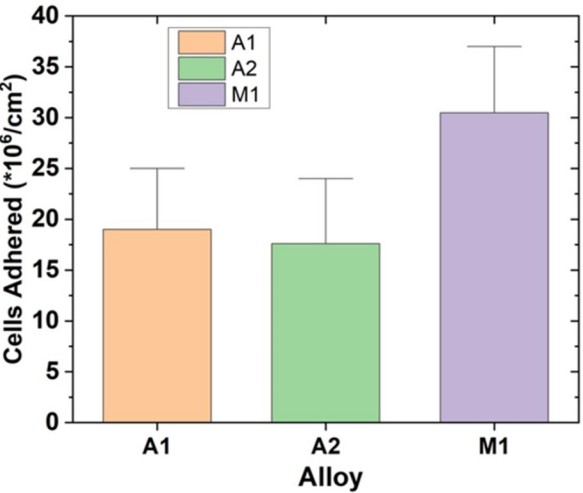

**Fig 5. Enumerated cell density on each alloy after 24 hrs of immersion.**

biofilm, which could contribute to its better corrosion resistance compared to alloys A1 and M1 corroborating the trends observed in pH and OCP experiments as discussed in Section 4.1 and 4.2.

### 3.4 SEM-EDX and imaging results

SEM imagery of bacterial adhesion in biotic conditions can be seen in Fig 6.

Fig 6A and 6B show the SEM images of alloy A1 after 7 and 14 days of immersion, respectively. In these images, *P.Aeruginosa* cells appear as dark, rod-shaped structures approximately 2–3 μm in length. After 7 days (Fig 6A), a relatively sparse distribution of bacterial cells can be observed on the surface of alloy A1. However, after 14 days (Fig 6B), significantly larger clumps of cells are visible, indicating the progression of biofilm formation over time.

Similarly, Fig 6C and 6D depict the SEM images of alloy A2 after 7 and 14 days of immersion. As with alloy A1, the bacterial cell density on alloy A2 increased from day 7 to day 14, demonstrating the growth of the biofilm. However, the biofilm on alloy A2 appears to be very sparsely distributed compared to that on alloy A1, which may be attributed to the higher chromium content in alloy A2 (30.7 wt% Cr) promoting the formation of a more stable and protective passive film [25,26].

Fig 6E and 6F show the SEM images of alloy M1 after 7 and 14 days of immersion. The biofilm growth on alloy M1 is more pronounced compared to alloys A1 and A2, with larger clusters of bacterial cells visible after both 7 and 14 days. This observation is consistent with the adhesion results discussed in Section 4.3, where alloy M1 exhibited a higher initial bacterial attachment due to its martensitic microstructure.

Optical microscopy was used to examine the microstructure of the alloys before and after corrosion. Fig 7. presents the as-deposited and post-14-day biotic corrosion surfaces of alloys M1, A1, and A2, respectively. In the as-deposited state, all three alloys display a characteristic microstructure consisting of a matrix (austenitic for A1 and A2, martensitic for M1) and dispersed carbides.

After 14 days of exposure to P. aeruginosa, the corrosion damage on the alloy surfaces is evident. Alloy M1 (Fig 7B) shows a higher number of pits compared to its as-deposited state (Fig 7A), with pits forming on both the matrix and the carbides. Similarly, alloy A1 (Fig 7D)

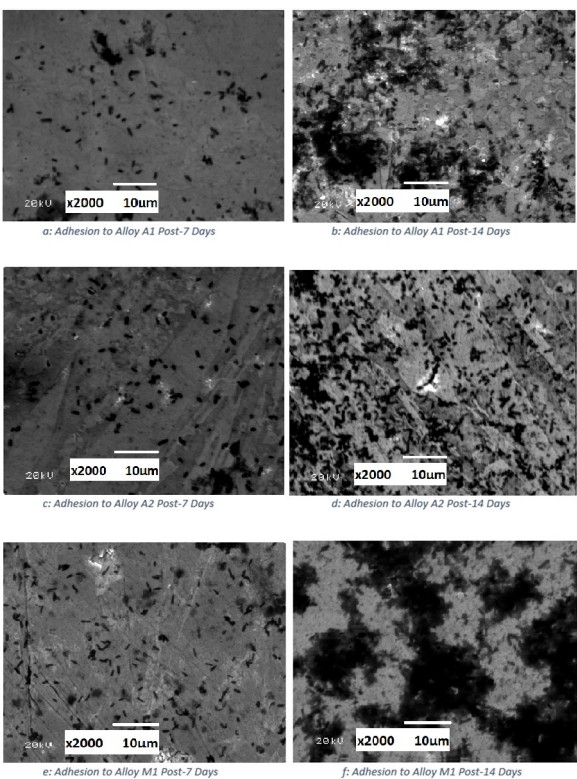

**Fig 6. SEM surfaces of bacterial adhesion after 7 and 14 days.** (a) Alloy A1 after 7 days (b) Alloy A1 after 14 days (c) Alloy A2 after 7 Days (d) Alloy A2 after 14 days (e)Alloy M1 after 7 days (f) Alloy M1 after 14 days.

exhibits significant pitting corrosion compared to its as-deposited surface (Fig 7C). In contrast, alloy A2 (Fig 7F) shows the least corrosion damage among the three alloys, with fewer and smaller pits compared to its as-deposited state (Fig 7E). This observation further supports the superior corrosion resistance of alloy A2 due to its higher chromium content.

Analysis of surface imagery can be undertaken using imageJ software. This was done using three equally sized surfaces for each sample, with indicative surface images shown in Fig 7. Table 3 presents the quantitative analysis of the pits formed on the alloy surfaces after 14 days of biotic corrosion. Alloy A2 had the lowest number of pits (153) and the smallest average pit size (projected area 0.74 $\mu m^2$), while alloys A1 and M1 had 239 and 269 pits, with average pit sizes (projected area) of 3.04 $\mu m^2$ and 1.27 $\mu m^2$, respectively. These results demonstrate the better corrosion resistance of alloy A2 compared to alloys A1 and M1.

Figs 8–10 show the surfaces of alloys M1, A1, and A2 after 14 days of exposure to abiotic conditions (without P. aeruginosa). In contrast to the biotic conditions, the corrosion in abiotic conditions appears to occur primarily on the matrix, with alloy A1 (Fig 9). showing almost complete removal of the matrix compared to alloys M1 (Fig 8) and A2 (Fig 10). This difference in corrosion behaviour between biotic and abiotic conditions highlights the significant impact of *P. Aeruginosa* on the corrosion mechanism, with the bacteria inducing localized pitting corrosion rather than the general matrix corrosion observed in abiotic conditions.

### 3.5 EIS results

Electrochemical Impedance Spectroscopy results can be fit to different circuit models using the ZSimpwin software. For coatings, this is expected to be a model equivalent to

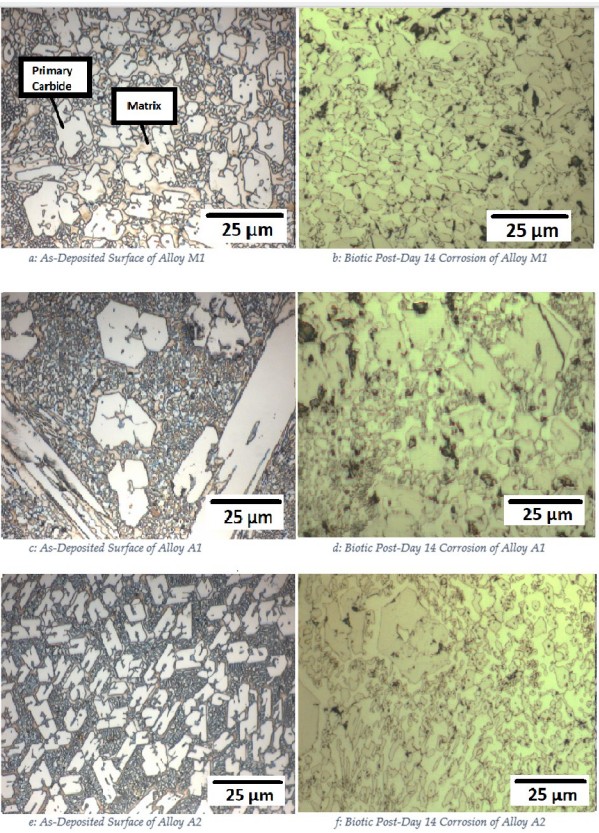

**Fig 7. SEM surfaces under as deposited and post-corrosion states.** (a) As-Deposited Surface of Alloy M1 (b) Post-Corrosion Surface of Alloy M1 after 14 Days Biotic Conditions (c) As-Deposited Surface of Alloy A1 (d) Post-Corrosion Surface of Alloy A1 after 14 Days Biotic Conditions (e) As-Deposited Surface of Alloy A2 (f) Post-Corrosion Surface of Alloy A2 after 14 Days Biotic Conditions.

$R_s[C_c[R_{po}[C_{cor}R_{cor}]$, where $R_{sol}$ indicates the resistance of the solution, $R_{po}$ indicates porous resistance (ie. of the coating), $C_c$ indicates coating capacitance, $C_{cor}$ indicates the double layer capacitance and $R_{cor}$ indicates charge transfer resistance [19]. This general model is as shown in Fig 11.

In terms of practical relevance, it is hypothesised that the system behaves in this manner for the abiotic results as the surface layer carbides acts as its own pseudo-coating, as seen due to the reliability of fit during abiotic conditions; however, it is also possible that this was caused by the formation of other layers such as chromium oxides. This model was also found to fit more reliably when modelling the capacitor elements instead as constant phase elements, which may represent non-homogenous in the system such as from being an imperfect coating.

During the conditions with bacteria, however, the system was found to be modelled better using $R_s[Q_b[R_b[Q_c(R_{po}(Q_{cor}R_{cor}))]]$, where $Q_b$ and $R_b$ represent the constant phase element

**Table 3. Enumerated pits on the surface of corroded alloys post-14 days in biotic conditions.**

| Sample | Total Pits | Largest Pit ($\mu m^2$) | Average Pit ($\mu m^2$) | SD ($\mu m^2$) |
|---|---|---|---|---|
| A1 | 239 | 57.45 | 3.04 | 6.42 |
| A2 | 153 | 8.39 | 0.74 | 1.14 |
| M1 | 269 | 21.75 | 1.27 | 2.64 |

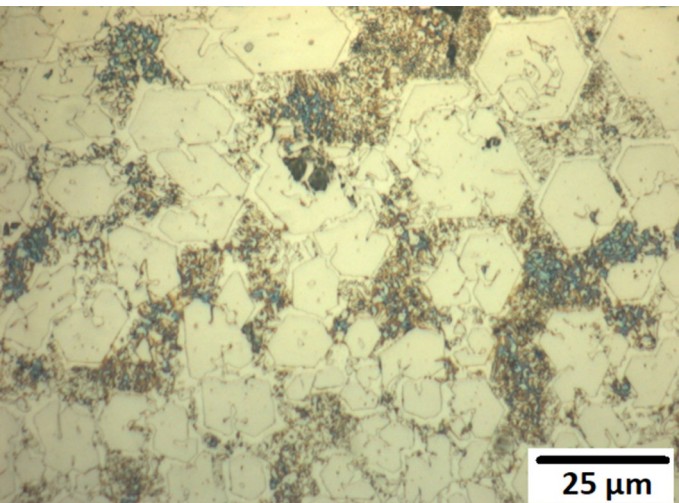

**Fig 8. Abiotic post-day 14 corrosion of Alloy M1.**

and resistance of the bacteria respectively. This model is as shown in Fig 12 and is likely due to the development of a multilayer coating with the formation of bacteria layer on the surface. Interpretation of this model towards physical conditions is assumed similar to work by Policastro regarding multilayer physical systems [30]; however, caution should still be exercised when comparing constants between different models, due to potential differences between assumed and real mechanisms. EIS results for both biotic and abiotic conditions are as shown in Figs 13–15.

Chi-squared values for all models fit were less than $10^{-3}$, indicating high quality of fit. From the Nyquist plots for Alloy A1 and M1 in Fig 13. it can be observed that the capacitive arc size increased significantly when comparing the abiotic conditions. This may be attributed to growth of the biofilm increasing the overall resistance of the system, which can also be corroborated from the increase of at least an order of magnitude of $R_{cor}$ for these results (eg. 2.15 $k\Omega cm^2$ in abiotic conditions for Alloy M1 Day 14 compared to 15.8 $k\Omega cm^2$ in biotic

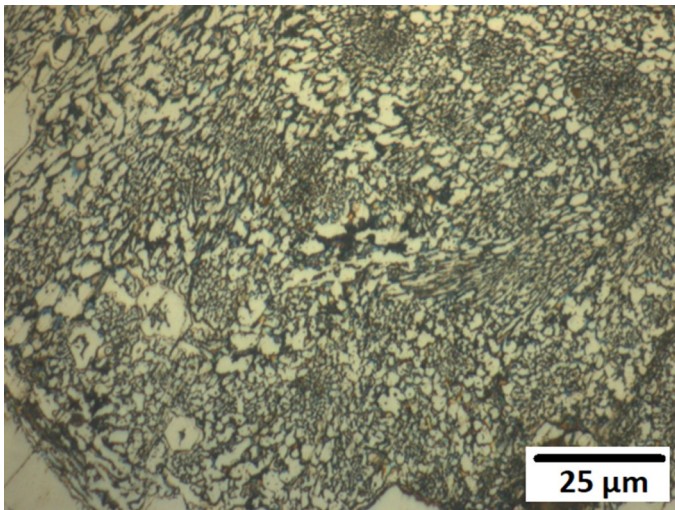

**Fig 9. Abiotic post-day 14 corrosion of Alloy A1.**

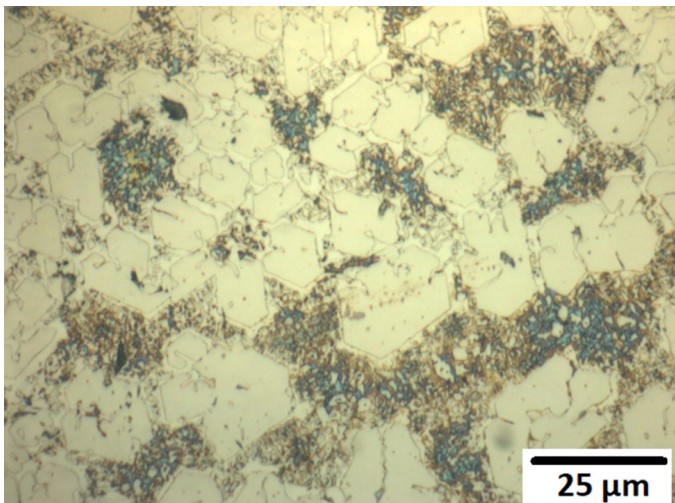

**Fig 10. Abiotic post-day 14 corrosion of Alloy A2.**

conditions for Alloy M1 Day 14). By Day 14, Alloy A2 had the highest $R_{cor}$ of 206 k$\Omega$cm$^2$ compared to Alloy A1 (7.02 k$\Omega$cm$^2$) and Alloy M1 (15.8 k$\Omega$cm$^2$). This supports the optical image that showed Alloy A2 having the most corrosion resistance due to having the lowest number and average size of pits. It could be suggested that that this is due to the higher chromium content in Alloy A2 affecting the growth of *P.Aeruginosa*, given the lower capacitance of bacteria on Alloy A2 Day 14 compared to Alloy A1 and M1 (4.52 x 10$^{-7}$ for Alloy A2 vs 4.73 x 10$^{-5}$ and 6.72 x 10$^{-5}$ for Alloy A1 and M1 respectively); however, this is not corroborated by the adhesion results in Fig 5. which indicated that the adhesion of bacteria on Alloy A2 was approximately equal to that of Alloy A1.In addition to the Nyquist plots, the Bode plots provide valuable information about the corrosion behaviour and surface properties of the HCWI alloys in the presence and absence of *P. Aeruginosa*. Bode plots are graphical representations of the impedance magnitude (|Z|) and phase angle (θ) as a function of frequency. In this study, the

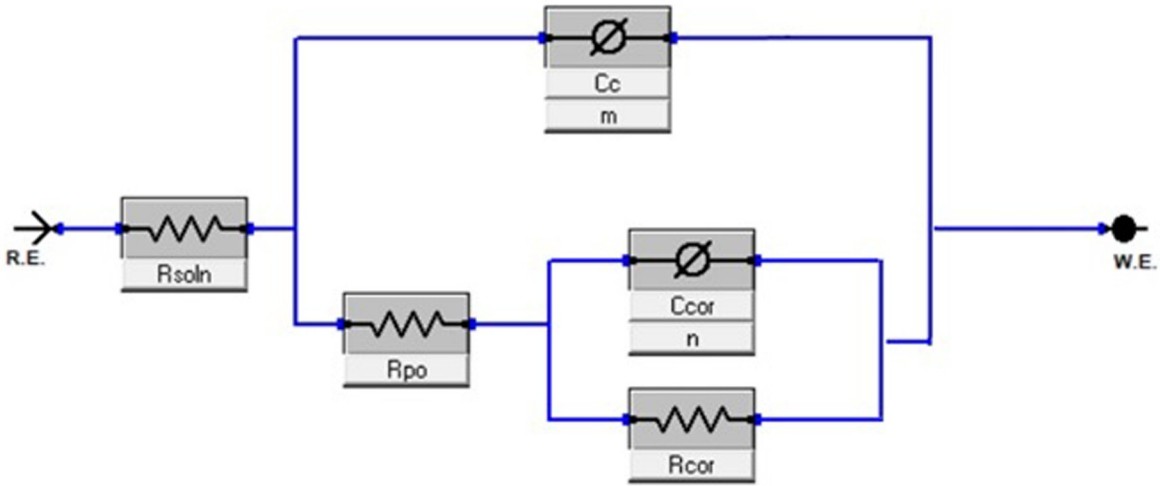

**Fig 11. Equivalent circuit model of Abiotic system in the study.**

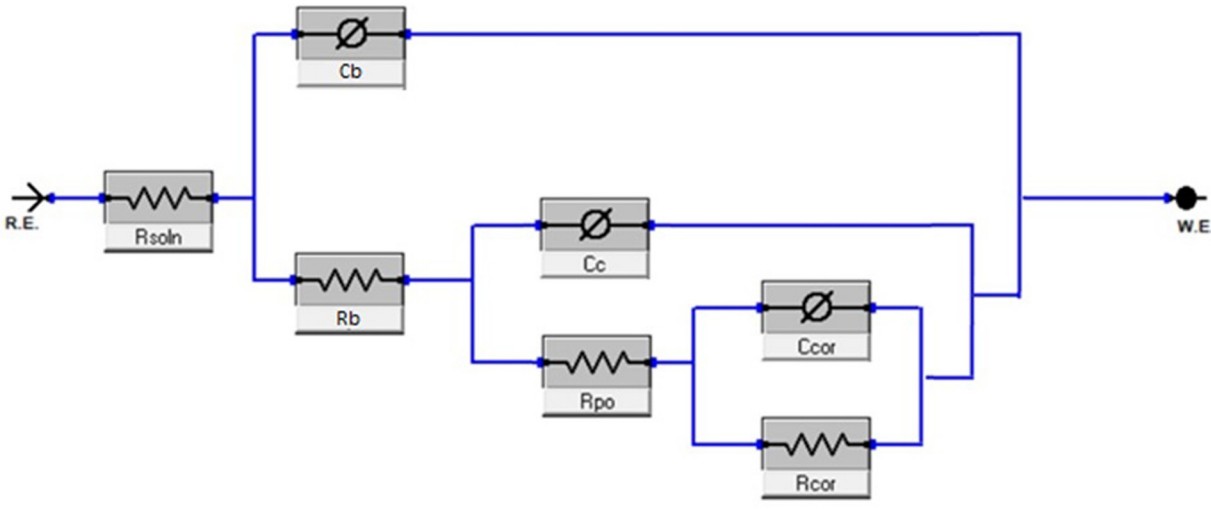

**Fig 12. Alternative equivalent circuit model of Biotic system in the study.**

Bode plots for alloys M1, A1, and A2 under biotic conditions after 7 and 14 days of immersion in the artificial seawater medium are presented in Figs 14 and 15 respectively.

Analysing the Bode plots for alloy A2 for days 7 and 14 (Fig 14), a clear difference can be observed under biotic conditions, i.e. the impedance magnitude at low frequencies (left side of the spectrum) is significantly higher than for A1 and M1, suggesting an enhanced corrosion resistance due to the formation of a biofilm on the alloy surface. The phase angle (θ) as observed in Fig 15 also shows a shift towards higher frequencies in the biotic conditions, which is indicative of the formation of a protective layer (biofilm) on the alloy surface [31]. This observation corroborates the findings from the Nyquist plots, confirming that alloy A2 exhibits the best corrosion resistance among the three alloys in the presence of *P.Aeruginosa*.

Comparing the Bode plots for alloy A2 at different exposure times (7 and 14 days) under biotic conditions reveals that the impedance magnitude at low frequencies also increases with longer exposure time. This trend suggests that the corrosion resistance of the alloys improves

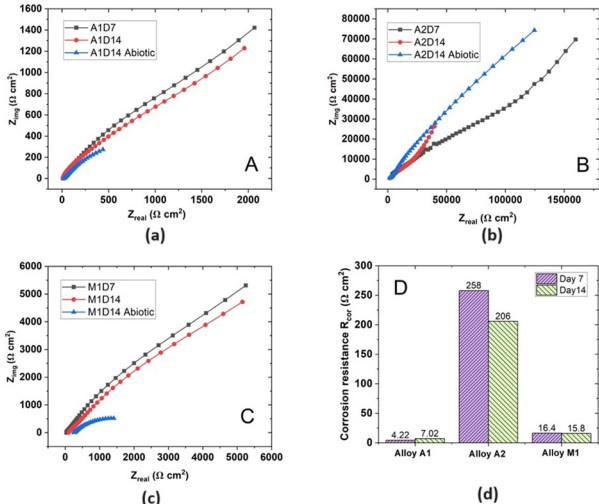

**Fig 13. Nyquist plots.** (a) Alloy A1 (b) Alloy A2 (c) Alloy M1 and (d) Corrosion resistance parameter of alloys.

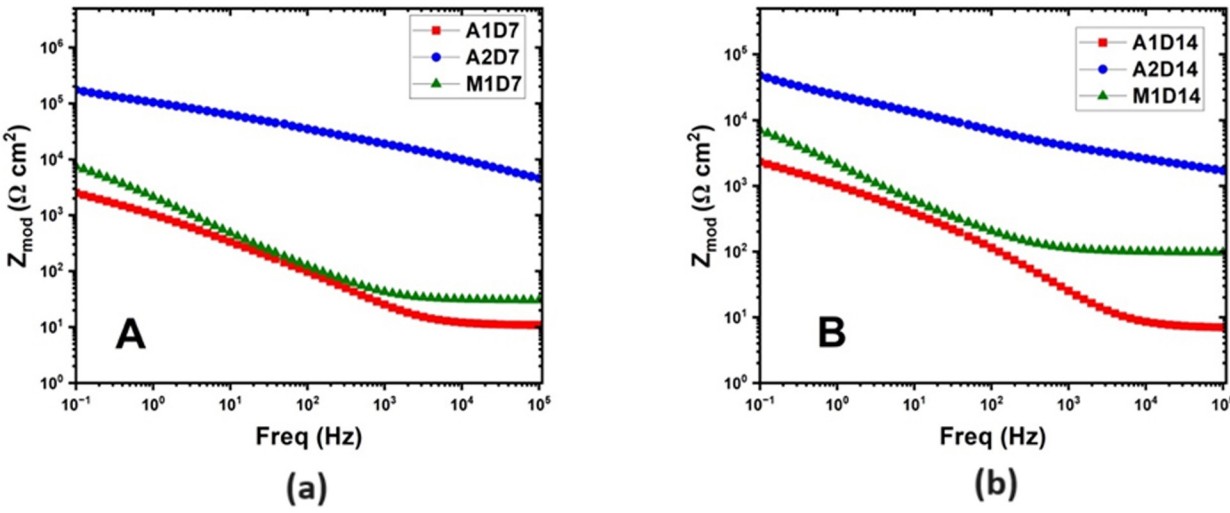

**Fig 14. Bode plots with Zmod.** (a) On Day 7 (b) On Day 14 biotic conditions.

as the biofilm matures and becomes more stable over time [19]. In the abiotic condition (Fig 16), the impedance magnitude (|Z|) at low frequencies is lower than the biotic conditions for A2 alloys, clearly indicating a lower corrosion resistance than the alloy with bacterial film.

The presence of multiple time constants (i.e., multiple peaks in the phase angle plot) suggests the existence of different interfaces and layers on the alloy surface, such as the passive film, biofilm, and corrosion products [32]. The equivalent circuit models presented in Figs 11 and 12 account for these different layers and interfaces, enabling a more accurate interpretation of the corrosion behaviour of the alloys under abiotic and biotic conditions.

## 4. Discussion

Biofilm formation on the surface of alloys is the first step in bacteria colonising the surface of materials. In aqueous environments, chromium is only stable in +III or +VI oxidation states

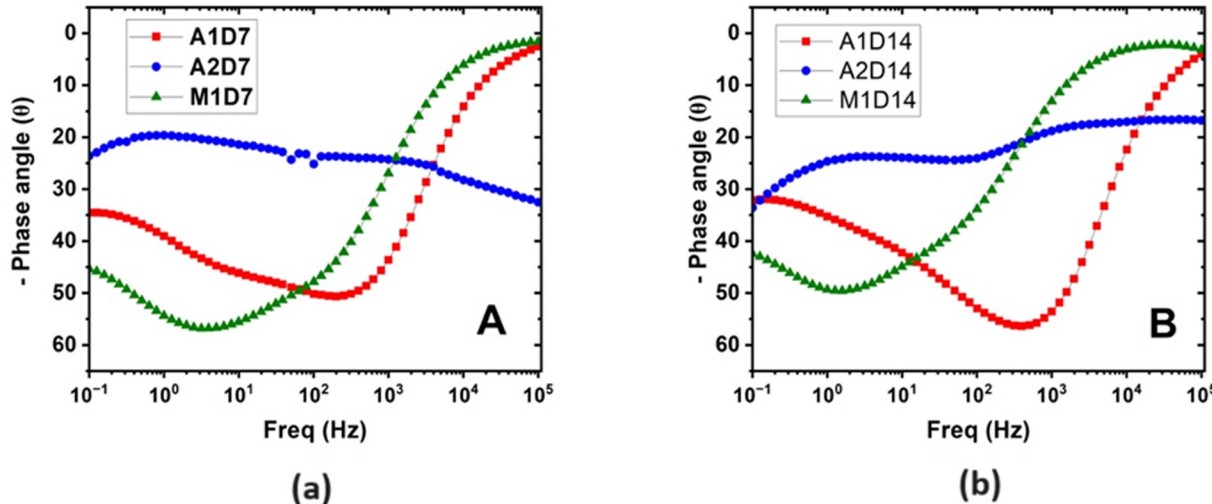

**Fig 15. Bode plot–phase angle Vs frequency.** (a) On Day 7 (b) On Day 14 biotic conditions.

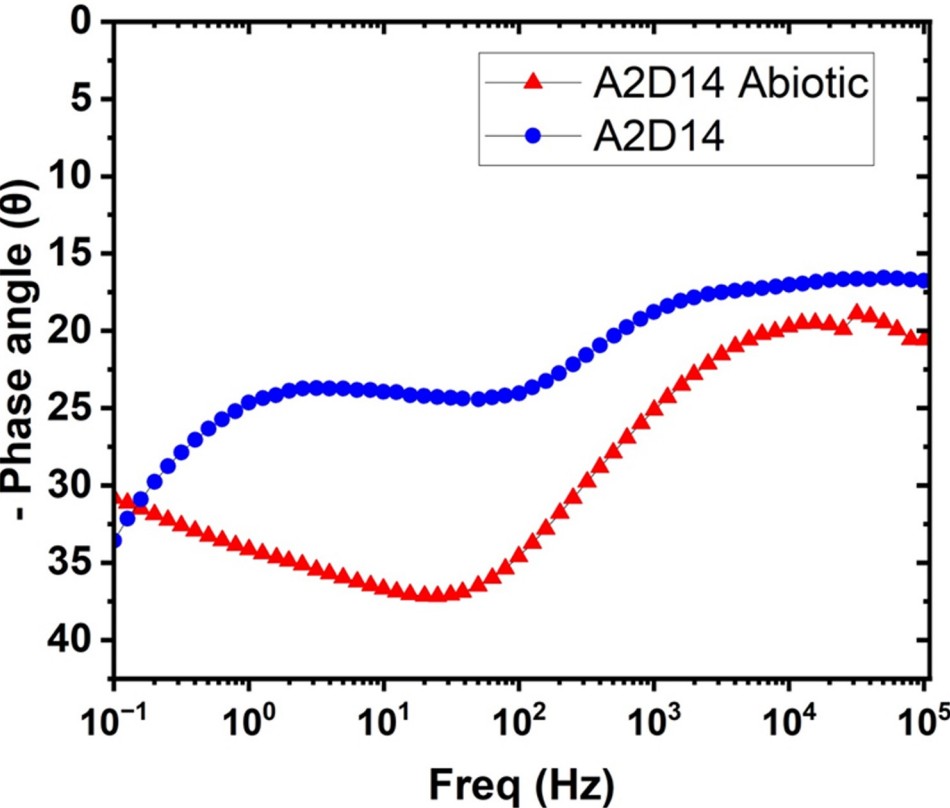

**Fig 16. Bode plot (Zmod vs Freq) for day 14 Biotic vs Abiotic conditions.**

[33]. Both Cr(III) and Cr(VI) are noted to be toxic to many microbial species due to buildup within cell cytoplasm [34]. However, *P.Aeruginosa* has the ability to be somewhat resistant towards these effects due to its ability to sequester Cr(VI) ions outside of its cell, reducing toxic buildup [35].

Adhesion of bacteria on the surface of all HCWI alloys was observed in SEM results after a week's immersion, whilst greater biofilm formation was observed at the end of the two-week exposure. This aligns with the general *P.Aeruginosa* biofilm formation observed for stainless steel, despite the less chromium content than HCWI. In the case of stainless steel, biofilm is typically described as patchy formations and growing thicker over time [13]. With the *P.Aeruginosa* biofilm, steel tends to corrode in three different ways [11]:

1. Formation of differential aeration cells, where a lack of oxygen beneath the biofilm creates an anodic area, whilst a cathode simultaneously forms elsewhere on the non-biofilm covered surface, where oxygen availability is more. These two areas are electrochemically connected to form a corrosion cell, accelerating the localised deterioration of material beneath the anode such as in the form of pits.

2. Interaction of steel with exo polymeric substances (EPS), which releases protons from carboxylic groups within the slime biofilm EPS, in turn fuelling energy for bacterial growth, as well as increasing acidity. Increased acidity may also exacerbate acid attack beneath the biofilm. [11]

3. Sidophores from P.Aeruginosa, which are complex low molecular weight compounds that can solubilise iron from steel, both increasing corrosion as well as aiding the growth and activity of the bacteria. [11]

In abiotic conditions, the corrosion occurred primarily on the matrix, with Alloy A1 showing almost complete removal of the matrix compared to Alloys A2 and M1 after 14 days (Figs 8–10). This general corrosion behaviour can be attributed to the galvanic effect between the matrix and the carbides, which is exacerbated by the attack of chloride ions, as noted in previous studies on high chromium-content alloys [36,37].

In contrast, under biotic conditions, localized pitting was the primary form of corrosion, as seen in Fig 7. This change in corrosion behaviour can be explained by the formation of differential aeration cells caused by the *P.Aeruginosa* biofilm, which was identified as the main corrosion mechanism. The biofilm creates localized differences in oxygen availability, leading to the establishment of anodic regions beneath the biofilm and cathodic regions on the exposed surface [11]. Additionally, the biofilm may have provided some protection against uniform chloride attack, contributing to the shift from general corrosion to localized pitting [38].

Despite the differences in corrosion mechanisms, the abiotic conditions generally resulted in more severe corrosion compared to the biotic conditions. This is evidenced by the lower $R_{cor}$ values obtained from EIS measurements in abiotic conditions, indicating a lower resistance to corrosion (Table 4). For example, on Day 14, the $R_{cor}$ values for Alloys A1, A2, and M1 increased from 1.72, 20.6, and 2.15 kΩcm$^2$ in abiotic conditions to 7.02, 206, and 15.8 kΩcm$^2$ in biotic conditions, respectively. Furthermore, the more extensive matrix removal observed in abiotic conditions (particularly for Alloy A1) may lead to poorer support for the carbides, potentially resulting in more severe wear-corrosion synergy, although this aspect was not investigated in the current study.

Comparing the corrosion resistance of the HCWI alloys in this study to that of stainless steels in the literature reveals some interesting observations. Alloys A1, A2, and M1 generally exhibited higher R_cor values (7.02, 206, and 15.8 kΩcm$^2$, respectively) after 14 days of exposure to *P.Aeruginosa* compared to the reported value of 1.83 kΩ for 304 stainless steel exposed to Pseudomonas NCIMB 2021 bacteria for the same duration [24]. However, 2304 duplex stainless steel (2304DSS) was found to have an R_cor of ~200 kΩcm$^2$ after 14 days of exposure to *P.Aeruginosa* [20], which is comparable to the value obtained for Alloy A2 in this study, despite 2304DSS having ~6% less chromium content. This suggests that other alloying elements present in 2304DSS, such as Mo and N, may also contribute to its corrosion resistance, particularly against pitting [39].

Similarly, the decay fit analysis of the OCP results provides additional insights into the initial corrosion behaviour of the HCWI alloys. The exponential decay function is given by:

$$OCP(t) = OCP\_0 + A*exp(-t/\tau)$$

where OCP(t) is the OCP value at time t, OCP_0 is the initial OCP value, A is the amplitude of the decay, and τ is the decay time constant. The decay rate can be calculated as 1/τ.

**Table 4. Electrochemical impedance parameters obtained with abiotic conditions.**

| | $R_{soln}$ (Ωcm$^2$) | $Q_{coating}$ (Ω$^{-1}$cm$^{-2}$s$^n$) | n1 | $R_{pore}$ (Ωcm$^2$) | $Q_{cor}$ (Ω$^{-1}$cm$^{-2}$s$^n$) | n2 | $R_{cor}$ (kΩcm$^2$) | Chi$^2$ |
|---|---|---|---|---|---|---|---|---|
| A1(Day 14) | 0.0711 | 6.16E-09 | 1.00 | 19.85 | 0.001945 | 0.5013 | 1.72 | 4.93E-05 |
| A2(Day 14) | 86.07 | 6.37E-06 | 0.322 | 5263 | 1.38E-06 | 0.5973 | 20.6 | 9.33E-05 |
| M1(Day 14) | 20.45 | 4.36E-09 | 0.880 | 240.1 | 0.000525 | 0.5677 | 2.15 | 2.31E-05 |

The decay fit results show that Alloy M1 has the highest decay rate (2.5 day$^{-1}$), followed by Alloy A1 (2.0 day$^{-1}$) and Alloy A2 (1.0 day$^{-1}$). The higher decay rate indicates a more rapid decrease in OCP during the initial stages of exposure to the corrosive environment, which can be attributed to the faster formation of the biofilm and the establishment of the corrosion cells [24].

Alloy A2, with the lowest decay rate, demonstrates a slower decrease in OCP, suggesting a more gradual formation of the biofilm and corrosion cells. This slower decay rate can be linked to the higher corrosion resistance of Alloy A2, as evidenced by its superior performance in the EIS results (higher charge transfer resistance and lower biofilm capacitance) and surface analysis (lower pit density and size) (Sections 4.4 and 4.5).

The higher decay rates of Alloys A1 and M1 indicate a more rapid establishment of the corrosion cells, which can lead to increased corrosion rates compared to Alloy A2. This is consistent with the lower charge transfer resistance and higher biofilm capacitance observed for these alloys in the EIS results (Table 5).

Given these results, mechanisms of corrosion can be proposed for Alloys A1 and A2 once biofilm formation has established itself on the surface on the surface. For both alloys it is suggested that a passive chromium oxide film developed on the surface. This is due to suitability when modelling equivalent circuit elements, as seen Fig 11. depicting the EIS double layer model within abiotic conditions. However, as stated previously, Alloy A1 had a less stable biofilm and passive film. This may be attributable to Alloy A1 having less chromium content than Alloy A2 (22% vs 30.7%), allowing Alloy A2 to form a more stable passive film. Lower biofilm capacitance for Alloy A2 compared to Alloy A1 (4.52 x 10$^{-7}$ vs 4.73 x 10$^{-5}$ $\Omega^{-1}$cm$^{-2}$s$^n$ on Day 14) and slightly lower adhesion for Alloy A2 (Fig 5). suggests that the despite being more stable, the biofilm for Alloy A2 was more compact and less porous. This reduced porosity of biofilm, along with stable passive film, is expected to reduce the transfer of chlorine ions within the solution through such film layers. As a result, Alloy A2 exhibits better corrosion resistance due to environmental protection provided by these films. An illustration of this proposed mechanism is as shown in Fig 17.

An interesting observation from this study is that Alloy M1, with its martensitic microstructure, exhibited slightly better corrosion resistance than Alloy A1, which has an austenitic microstructure, despite their similar chromium contents (21 wt% and 22 wt% Cr, respectively). This is evidenced by the smaller average pit size and higher $R_{cor}$ value of Alloy M1 compared to Alloy A1 after 14 days of biotic corrosion. In specific, after 14 days of exposure to *P. Aeruginosa*, Alloy M1 exhibited a higher charge transfer resistance ($R_{cor}$ = 15.8 k$\Omega$cm$^2$) compared to Alloy A1 (R_cor = 7.02 k$\Omega$cm$^2$), indicating better corrosion resistance (Table 5). Corroborating this observation, the Bode plot (Fig 14). show that Alloy M1 had higher impedance values at low frequencies compared to Alloy A1, further confirming its superior corrosion resistance. Similarly, optical microscopy images of the corroded surfaces after 14 days of biotic exposure (Fig 7). reveal that Alloy M1 had a lower average pit size (1.27 $\mu$m$^2$) compared to

**Table 5. Electrochemical impedance parameters obtained with biotic conditions.**

| | $R_{soln}$ ($\Omega$cm$^2$) | $Q_{bacteria}$ ($\Omega^{-1}$cm$^{-2}$s$^n$) | n1 | $R_{bacteria}$ ($\Omega$cm$^2$) | $Q_{coating}$ ($\Omega^{-1}$cm$^{-2}$s$^n$) | n2 | $R_{pore}$ ($\Omega$cm$^2$) | $Q_{cor}$ ($\Omega^{-1}$cm$^{-2}$s$^n$) | n3 | $R_{cor}$ (k$\Omega$cm$^2$) | Chi$^2$ |
|---|---|---|---|---|---|---|---|---|---|---|---|
| A1 (Day 7) | 10.7 | 4.37E-05 | 0.805 | 135.9 | 0.000249 | 0.540 | 1.21E+04 | 0.000323 | 0.691 | 4.22 | 4.99E-06 |
| A2 (Day 7) | 724.3 | 1.46E-07 | 0.540 | 11100 | 3.44E-06 | 0.334 | 1.92E+06 | 7.09E-06 | 0.686 | 258 | 1.30E-04 |
| M1 (Day 7) | 30.0 | 4.88E-05 | 0.784 | 291.2 | 7.08E-05 | 6.69E-01 | 5.34E+04 | 0.000116 | 0.724 | 16.4 | 1.04E-05 |
| A1 (Day 14) | 9.21 | 4.73E-05 | 0.791 | 159.2 | 0.000357 | 0.488 | 1.10E+04 | 0.000203 | 0.795 | 7.02 | 3.28E-05 |
| A2 (Day 14) | 47.5 | 4.52E-07 | 0.567 | 1919 | 1.13E-05 | 0.344 | 8.81E+05 | 1.57E-05 | 1.00 | 206 | 1.88E-05 |
| M1 (Day 14) | 96.6 | 6.72E-05 | 0.712 | 787.3 | 6.75E-05 | 0.6743 | 4.12E+04 | 0.9512 | 0.742 | 15.8 | 1.21E-04 |

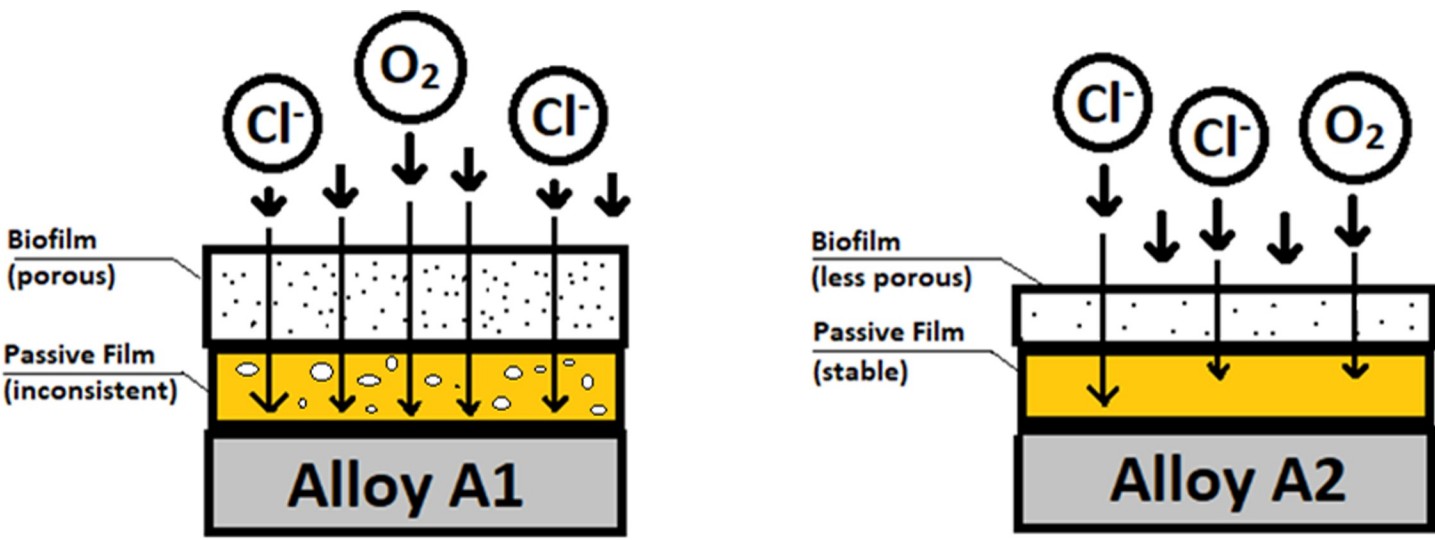

**Fig 17. Proposed transfer mechanism for Alloys A1 and A2.**

Alloy A1 (3.04 μm$^2$), despite having a similar number of pits (269 and 239, respectively) (Table 3). The smaller pit size on Alloy M1 suggests that it has better resistance to localized corrosion compared to Alloy A1. Fluorescence microscopy results also show that Alloy M1 had a significantly higher bacterial adhesion (32 × 10$^6$ cells/cm$^2$) compared to Alloy A1 (17 × 10$^6$ cells/cm$^2$) after 24 hours of immersion in the test medium. The higher bacterial adhesion on Alloy M1 can be attributed to its martensitic microstructure, which has a higher density of surface dislocations that can act as preferential sites for bacterial attachment [28].

Despite the higher bacterial adhesion on Alloy M1, it exhibited better corrosion resistance than Alloy A1. This suggests that the corrosion mechanism is not solely dependent on the initial bacterial attachment. The superior corrosion resistance of Alloy M1 compared to Alloy A1 may be due to the formation of a more homogeneous and protective biofilm, which could limit the cathodic area available for the differential aeration cell mechanism [11]. The better performance of Alloy M1 compared to Alloy A1 suggests that microstructure plays a significant role in determining the corrosion behaviour of HCWI alloys under MIC conditions, and that the relationship between bacterial adhesion and corrosion resistance is not always straightforward when the alloy's chromium content is similar.

Overall, the superior corrosion resistance of Alloy A2 compared to Alloys A1 and M1 can be attributed to its higher chromium content (30.7 wt% Cr). The higher chromium content promotes the formation of a more stable and protective passive film, which is crucial for corrosion resistance in high chromium iron alloys. [14,25]

## 5. Conclusions

In this study, the Microbiologically Influenced Corrosion (MIC) resistance of three High Chromium White Iron (HCWI) alloys with varying chromium contents (22 wt%, 30.7 wt%, and 21 wt%) was investigated in the presence of Pseudomonas aeruginosa, a common bacterial species associated with MIC. The comprehensive evaluation of the corrosion behaviour using electrochemical techniques, surface analysis, and microscopy provided valuable insights into the role of chromium content and the underlying corrosion mechanisms. Electrochemical Impedance Spectroscopy (EIS) results, interpreted using equivalent circuit models and Bode plots, clearly demonstrated the superior MIC resistance of alloy A2 (30.7 wt% Cr) compared to

alloys A1 (22 wt% Cr) and M1 (21 wt% Cr). The higher chromium content in alloy A2 was found to promote the formation of a more stable and protective passive film, as well as the development of a more compact and less permeable biofilm. These factors collectively contributed to the enhanced corrosion resistance of alloy A2, as evidenced by its higher charge transfer resistance, lower biofilm capacitance, and higher impedance values at low frequencies. Surface analysis techniques, including Scanning Electron Microscopy (SEM) and optical microscopy, corroborated the electrochemical findings, revealing that alloy A2 had the lowest pit density and size after 14 days of exposure to the bacterial environment. The schematic representation of the corrosion mechanism further elucidated the critical role of chromium content in the stability of the passive film and the permeability of the biofilm, which ultimately influenced the transport of corrosive species to the metal surface and the overall corrosion resistance. The findings of this study have significant implications for the design and selection of materials for applications prone to microbial corrosion. The superior performance of alloy A2 highlights the potential for developing HCWI alloys with enhanced MIC resistance by optimizing the chromium content.

## Supporting information

**S1 File.**
(ZIP)

## Author Contributions

**Conceptualization:** Cedric Tan.

**Data curation:** Cedric Tan.

**Formal analysis:** Cedric Tan, Naveen Kumar Elumalai.

**Investigation:** Cedric Tan, Kannoorpatti Narayanan Krishnan.

**Methodology:** Cedric Tan, Kannoorpatti Narayanan Krishnan.

**Supervision:** Naveen Kumar Elumalai, Kannoorpatti Narayanan Krishnan.

**Visualization:** Kannoorpatti Narayanan Krishnan.

**Writing – original draft:** Cedric Tan.

**Writing – review & editing:** Cedric Tan, Naveen Kumar Elumalai, Kannoorpatti Narayanan Krishnan.

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
