## [Decision Letter · Decision Letter 0]

30 May 2024

PONE-D-24-18779Microbiologically Influenced Corrosion by Pseudomonas Aeruginosa on High Chromium White IronPLOS ONE

Dear Dr. Krishnan,

Thank you for submitting your manuscript to PLOS ONE. After careful consideration, we feel that it has merit but does not fully meet PLOS ONE’s publication criteria as it currently stands. Therefore, we invite you to submit a revised version of the manuscript that addresses the points raised during the review process.

**ACADEMIC EDITOR: **Based on the Reviewers’ comments, please revise the proposed manuscript adequately. Be sure that all comments and suggestions are solved and provided. additionally, be sure that “reproducibility” be guaranteed and proved. This must obligatory be carried out by using duplicate and /or triplicate into those depicted results/data.Best regards ==============================

We look forward to receiving your revised manuscript.

Kind regards,

Wislei Riuper Osório

Academic Editor

PLOS ONE

Journal Requirements:

 "Australian Government RTP funding was provided as scholarship to Cedric Tan"

Additional Editor Comments:

Based on the Reviewers’ comments, please revise the proposed manuscript adequately.

Reviewers' comments:

Reviewer's Responses to Questions

**Comments to the Author**

1. Is the manuscript technically sound, and do the data support the conclusions?

Reviewer #1: Partly

Reviewer #2: Partly

2. Has the statistical analysis been performed appropriately and rigorously? 

Reviewer #1: N/A

Reviewer #2: Yes

3. Have the authors made all data underlying the findings in their manuscript fully available?

Reviewer #1: Yes

Reviewer #2: Yes

4. Is the manuscript presented in an intelligible fashion and written in standard English?

Reviewer #1: Yes

Reviewer #2: Yes

5. Review Comments to the Author

Reviewer #1: Tan et al., provide valuable insights into the MIC resistance of HCWI alloys and underscores the importance of chromium content in combating microbial corrosion. The comprehensive approach and detailed analysis make a strong case for the use of higher chromium-containing alloys in corrosive environments. Future research could build on these findings by exploring longer exposure periods, different microbial species, and real-world environmental conditions to further validate and expand upon these results. Overall, the study makes a significant contribution to the field of material science and performance of HCWI alloys in MIC-prone applications. However, there are still some concerns should be addressed, and strongly suggest the author improve the quality of SEM images.

Major:

1. What is the rational to study the specific chromium contents of the three HCWI alloys (22 wt% (A1), 30.7 wt% (A2), and 21 wt% (M1))? Contents that related to nature environment most should be addressed.

2. The only microorganism used was Pseudomonas aeruginosa, a common bacterial species associated with MIC. Did the author compared other microorganisms that might have similar outcomes? Or is there any combination of different microorganisms that have same effects?

3. Author showed that alloy A2 (30.7 wt% Cr) exhibited the best MIC resistance. How can author support the underlying mechanism?

4. What are the implications of this study for material selection in industries prone to microbial corrosion? Is there any comparisons between different tests?

5. The labels in figure 6 and 7 are unclear, it's very hard to read, can author change the label background or color of text to make it more clear?

6. Only one result can not strongly support the cell adhesion in figure 5, and this result lacks statistic analysis.

Reviewer #2: 1. You have performed the tests for 14 days under laboratory conditions. The test duration is too short and conditions not met in nature,

2.Based on the above item, I suggest to modify the title of the paper to highlight the fact that it is a pre-liminary, short-term research,

3. I also suggest that you address the shortcomings of this present research and how to address it in your future works.

4. I advise you to read recently published papers that shw that the term "biofilm" is not correct as it is neither 100% biological nor has a film structure. In clude this in your paper to increase professionals' awareness about the fact that Biofilm is not a correct way of addressing bacterial establishments.

5. Your manuscript needs English proof-reading.

6. PLOS authors have the option to publish the peer review history of their article (what does this mean?). If published, this will include your full peer review and any attached files.

Reviewer #1: No

Reviewer #2: **Yes: **Dr. Reza Javaherdashti

---

## [Author Response · Author response to Decision Letter 0]

5 Jun 2024

Response to reviewer comments are also as provided within "Response to Reviewers" file. Additionally, as per editor comments, the financial disclosure statement has been updated to:

Australian Government RTP funding was provided as scholarship to Cedric Tan. This funder had no role in study design, data collection and analysis, decision to publish, or preparation of the manuscript.

Reviewer 1:

1. What is the rational to study the specific chromium contents of the three HCWI alloys (22 wt% (A1), 30.7 wt% (A2), and 21 wt% (M1))? Contents that related to nature environment most should be addressed.

Hardfacing alloys were chosen based on their popularity in industry and are readily available from Australian welding consumable suppliers such as Cigweld [1]. WTIA (Welding Technology Institute of Australia) Technical Note 4 provides guidance to the selection of hardfacing alloys [2]. These alloys are described by Australian Standards under AS/NZS 2576:2005 [3]. Alloy A1 is categorised by AS 2576 under 23xx, and A2 as 24xx and M1 as 25xx. Specific alloys were chosen not only for chromium contents but also for their matrix structure. This is as Alloys A1 and A2 are austenitic, while Alloy M1displays martensitic microstructure [3]. These details are also discussed in the manuscript under Section 3.1.

These alloys are commonly used in mining, metallurgy, and wear-resistant applications, where they are exposed to a wide range of pH values and corrosive environments. By studying alloys with varying chromium contents, we aimed to investigate the impact of chromium on the formation of biofilms and its MIC resistance of HCWI alloys.

[1] Cigweld (2024), https://www.cigweld.com.au/range/welding-consumables/hardfacing-consumables/hardfacing-electrodes/, accessed 3/6/2024

[2] Gross, B. and W.T.I.o. Australia, The Industry Guide to Hardfacing for the Control of Wear. 1995: Welding Technology Institute of Australia.

[3] AS 2576-1982, Welding consumables for build-up and wear resistance - Classification system. 2023; Available from: https://infostore.saiglobal.com/

2. The only microorganism used was Pseudomonas aeruginosa, a common bacterial species associated with MIC. Did the author compared other microorganisms that might have similar outcomes? Or is there any combination of different microorganisms that have same effects?

We focused on Pseudomonas aeruginosa due to its prevalence in marine environments and its well-established role in MIC [4] [5]. While we did not compare the effects of other microorganisms or combinations of microorganisms in this study, we acknowledge that other microbial species may have similar or different impacts on the MIC resistance of HCWI alloys. Future research could explore the effects of different microorganisms, such as sulfate-reducing bacteria or acid-producing bacteria, as well as the synergistic effects of multiple microbial species on the corrosion behavior of HCWI alloys. We are planning to perform such investigations in our future work.

[4] Xiong, Y.-Q., et al., Influence of pH on adaptive resistance of Pseudomonas aeruginosa to aminoglycosides and their postantibiotic effects. Antimicrobial agents and chemotherapy, 1996. 40(1): p. 35-39.

[5] Abdolahi, A., et al., Microbially influenced corrosion of steels by Pseudomonas aeruginosa. Corrosion Reviews, 2014. 32(3-4): p. 129-141.

3. Author showed that alloy A2 (30.7 wt% Cr) exhibited the best MIC resistance. How can author support the underlying mechanism?

Microscopy results such as in Figure 5 indicated similar adhesion for Alloys A1 and A2, with significantly greater adhesion for Alloy M1. However, from the EIS results, the capacitance attributed to biofilm was lowest for Alloy A2 (4.52 x 10-7 Ω-1cm-2sn), suggesting a less permeable and more protective layer. It is speculated in the mechanism suggested in the paper that the biofilm is more compact and less porous. The proposed mechanism in Figure 17 illustrates how the stable passive film and compact biofilm on alloy A2 reduce the transfer of chloride ions and other corrosive species to the metal surface, resulting in enhanced corrosion resistance. 

4. What are the implications of this study for material selection in industries prone to microbial corrosion? Is there any comparisons between different tests?

Implications of this study show that having a more stable and protective passive film through the addition of chromium is beneficial for these alloys against MIC. In this investigation, we have used different tests such as microscopy and electrochemical characterisation to investigate the MIC of hardfacing alloys. Both these types of tests help to understand the extent of MIC and its correlation with alloy behaviour [6] [7].

Optical microscopy and SEM was used to compare surface morphology, whilst electrochemical tests was used to characterise charge transfer processes and corrosion mechanisms between the different alloys. Both of these methods were implemented in this work.

[6] Little, B.J. and J.S. Lee, Microbiologically Influenced Corrosion. 2007: Wiley.

[7] Little, B.J. et. al, ASM Handbook: Corrosion: Fundamentals, Testing, and Protection. Volume 13A. 2003: ASM International.

5. The labels in figure 6 and 7 are unclear, it's very hard to read, can author change the label background or color of text to make it more clear?

Figure texts have been amended to improve readability.

6. Only one result can not strongly support the cell adhesion in figure 5, and this result lacks statistic analysis.

Cell density was enumerated from 10 different randomly selected fields over the image surfaces, with three different image surfaces analysed. Error bars are also presented within Figure 5 which confirm significantly greater adhesion within Alloy M1. Higher concentration of cell clumps can also be observed within Alloy M1 on Figure 6 after 14 days compared to Alloy A1 and A2. This is mentioned in Section 3.4.

 

Reviewer 2:

 1. You have performed the tests for 14 days under laboratory conditions. The test duration is too short and conditions not met in nature/ 2.Based on the above item, I suggest to modify the title of the paper to highlight the fact that it is a pre-liminary, short-term research,

It is acknowledged that the work is performed under laboratory conditions only for 14 days. Hence, as suggested by the reviewer, the title has been modified to highlight that this investigation is preliminary. However, our investigation follows from literature for P.Aeruginosa that carried out MIC investigations over 14 days [1] [2] [3].

[1] Xu, D., J. Xia, E. Zhou, D. Zhang, H. Li, C. Yang, Q. Li, H. Lin, X. Li, and K. Yang, Accelerated corrosion of 2205 duplex stainless steel caused by marine aerobic Pseudomonas aeruginosa biofilm. Bioelectrochemistry, 2017. 113: p. 1-8.

[2] Zhou, E., H. Li, C. Yang, J. Wang, D. Xu, D. Zhang, and T. Gu, Accelerated corrosion of 2304 duplex stainless steel by marine Pseudomonas aeruginosa biofilm. International Biodeterioration & Biodegradation, 2018. 127: p. 1-9.

[3] Tran Thi Thuy, T., K. Kannoorpatti, A. Padovan, S. Thennadil, and N. Nguyen Dang, Effect of nickel on the adhesion and corrosion ability of Pseudomonas aeruginosa on stainless steel. Journal of Materials Engineering and Performance, 2019. 28: p. 5797-5805.

3. I also suggest that you address the shortcomings of this present research and how to address it in your future works.

Shortcomings of the current research include:

• In the investigation, tests were conducted only under neutral pH conditions. It would be interesting to carry out investigations under low and high pH solutions as this would simulate conditions in different mining environments. 

• Testing was conducted under stagnant electrolytic conditions. However, when these alloys are used in surfacing pumps, flow conditions are likely to affect the adhesion of bacteria and hence MIC. 

• The tests were conducted using P.Aeruginosa only. However, in field conditions in mining environments, colonies of different bacteria may affect each other and the MIC of alloys. 

• Addition of other elements such as tungsten and niobium to form complex carbides in the investigation of MIC would provide us with information on the effects of other elements. However, chromium was chosen as it is one of the cheaper elements to form carbides, as well as being known for its corrosion resistance. 

Addressing the above is being planned for future studies.

4. I advise you to read recently published papers that show that the term "biofilm" is not correct as it is neither 100% biological nor has a film structure. Include this in your paper to increase professionals' awareness about the fact that Biofilm is not a correct way of addressing bacterial establishments.

Manuscript has been amended to include the following text within the Introduction section acknowledging such terminology:

Javaherdashti argues that the term biofilm in MIC be replaced with the more suitable term “temenos” to emphasise upon the fact that under-‘biofilm’ conditions are far different from those of the bulk solution [4]. However, due to currently accepted usage and familiarity among the scientific community involved in MIC research, these layers are still referred to as “biofilms” in this paper

[4] Javaherdashti, R., Some thoughts about misconceptions surrounding the term ‘biofilm’. Corrosion Engineering, Science and Technology, 2020. 55(8): p. 681-684.

5. Your manuscript needs English proof-reading.

Manuscript has been further checked for errors to improve readability.

---

## [Editor Report · Decision Letter 1]

13 Jun 2024

Preliminary Study of Microbiologically Influenced Corrosion by Pseudomonas Aeruginosa on High Chromium White Iron

PONE-D-24-18779R1

Dear Dr. Krishnan,

We’re pleased to inform you that your manuscript has been judged scientifically suitable for publication and will be formally accepted for publication once it meets all outstanding technical requirements.

Kind regards,

Wislei Riuper Osório

Academic Editor

PLOS ONE

Additional Editor Comments (optional):

Based on the Reviewers’ responses, it is observed that majority questions are solved. It seem that its final publication is deserved.
---

## [Editor Report · Acceptance letter]

7 Jul 2024

PONE-D-24-18779R1 

PLOS ONE

Dear Dr. Krishnan, 

I'm pleased to inform you that your manuscript has been deemed suitable for publication in PLOS ONE. Congratulations! Your manuscript is now being handed over to our production team.

Kind regards, 

on behalf of

Dr. Wislei Riuper Osório 

Academic Editor

PLOS ONE